# Impaired Intestinal Permeability Assessed by Confocal Laser Endomicroscopy—A New Potential Therapeutic Target in Inflammatory Bowel Disease

**DOI:** 10.3390/diagnostics13071230

**Published:** 2023-03-24

**Authors:** Stefan Chiriac, Catalin Victor Sfarti, Horia Minea, Carol Stanciu, Camelia Cojocariu, Ana-Maria Singeap, Irina Girleanu, Tudor Cuciureanu, Oana Petrea, Laura Huiban, Cristina Maria Muzica, Sebastian Zenovia, Robert Nastasa, Remus Stafie, Adrian Rotaru, Ermina Stratina, Anca Trifan

**Affiliations:** 1Department of Gastroenterology, Grigore T. Popa University of Medicine and Pharmacy, 700115 Iasi, Romania; 2Institute of Gastroenterology and Hepatology, “St. Spiridon” University Hospital, 700111 Iasi, Romania

**Keywords:** intestinal permeability, inflammatory bowel disease, confocal laser endomicroscopy

## Abstract

Inflammatory bowel diseases (IBD) represent a global phenomenon, with a continuously rising prevalence. The strategies concerning IBD management are progressing from clinical monitorization to a targeted approach, and current therapies strive to reduce microscopic mucosal inflammation and stimulate repair of the epithelial barrier function. Intestinal permeability has recently been receiving increased attention, as evidence suggests that it could be related to disease activity in IBD. However, most investigations do not successfully provide adequate information regarding the morphological integrity of the intestinal barrier. In this review, we discuss the advantages of confocal laser endomicroscopy (CLE), which allows in vivo visualization of histological abnormalities and targeted optical biopsies in the setting of IBD. Additionally, CLE has been used to assess vascular permeability and epithelial barrier function that could correlate with prolonged clinical remission, increased resection-free survival, and lower hospitalization rates. Moreover, the dynamic evaluation of the functional characteristics of the intestinal barrier presents an advantage over the endoscopic examination as it has the potential to select patients at risk of relapses. Along with mucosal healing, histological or transmural remission, the recovery of the intestinal barrier function emerges as a possible target that could be included in the future therapeutic strategies for IBD.

## 1. Introduction

Inflammatory bowel diseases (IBD) are chronic conditions with an incompletely known pathogenic mechanism that includes a complex interaction between environmental factors, genetic profile, the immune system, and the microbiome of the patients. There are substantial differences in the epidemiological trend as well as clinical and evolutionary features. According to studies carried out from 1990 to 2016 in Europe, the highest prevalence of ulcerative colitis (UC) is reported in Norway (505 cases/100,000 inhabitants). In the case of Crohn’s disease (CD), the highest prevalence is reported in Germany (322 cases/100,000 inhabitants). At the opposite pole, the fewest cases of UC and CD are recorded in Bosnia and Herzegovina (43.1/100,000 inhabitants and 28.2/100,000 inhabitants, respectively) [1].

In the United States of America (USA), the prevalence of UC is 286 cases/100,000 inhabitants, and in Canada a CD prevalence of 368 cases/100,000 inhabitants. In the next decade, an increase of more than 0.5% in the prevalence of IBD is estimated in North America, which will result in approximately 4 million patients being affected by IBD [2]. 

Recent studies define IBD as a globally widespread disease with a steady upward trend in Europe and North America, which contrasts with the extremely low incidence rates in the developing countries of the Middle East, South Asia, and Sub-Saharan Africa [3,4]. However, after the 1990s, the adoption of a western lifestyle led to significant increases in cases of the disease in these regions. Moreover, immigrants who move to developed countries represent a unique population in which the risk for IBD evolves over subsequent generations to become comparable to that in the host country eventually [5,6]. Several studies conducted on large groups of Mexicans living in the US and Indians settled in Europe, the US, or the Middle East have shown large differences regarding incidence and prevalence compared to countries of origin, reflecting the impact of diet, pollution, or environmental factors on the epidemiology of the disease [7,8]. A recent Spanish study conducted by Gutierez et al. concluded that immigrant persons present a higher risk of early onset of IBD, more extraintestinal signs of the disease, and early introduction of biologic therapy than native people from Spain [9].

Moreover, IBD represents a significant economic burden, especially in North America and Western Europe, where total annual costs reach over 30 billion euros [10]. Selecting an appropriate therapeutic target in inflammatory bowel disease (IBD) is of paramount importance and has been considered a crucial step in the management of both UC and CD [11]. In the early years of treatment of IBD, only clinical response and clinical remission were used to guide therapy, but the need for more adequate control of the disease further led to the proposal of endoscopic healing (EH) as a long-term target [12,13]. EH, assessed using several endoscopic activity scores, such as the simple endoscopic score in CD (SES-CD) and Mayo endoscopic subscore (MES) in the case of UC, was associated with good outcomes in IBD. However, these scores did not provide information regarding mucosal healing and could not adequately assess the patency of the intestinal barrier. Recently, histologic remission emerged as a potential target, as it was shown to be associated with long-term remission in both UC and CD [14,15,16,17]. Unfortunately, the lack of standardization as well as the high costs involved in achieving this goal prevented the universal implementation of this target [11]. Confocal laser endomicroscopy (CLE) was introduced as a novel technique that allowed real-time evaluation of the intestinal mucosa, providing high-resolution in vivo microscopy images [18]. The usefulness of this method had been extensively assessed in several prospective studies and reviews proving its utility in the evaluation of mucosal healing and the prediction of IBD activity relapse [19,20,21,22]. While both CLE and histologic assessment could provide information on structural changes in the mucosa of IBD patients, only CLE evaluates certain functional features, such as the integrity of the intestinal mucosa. These discrete but clinically relevant changes were found to be associated with relapse in IBD patients without endoscopic activity [21]. The modern therapeutic approach to IBD, namely the treat-to-target strategy, requires the identification of adequate therapeutic targets to better stratify patients that present a high risk for disease activity as well as complications. We aimed to discuss the utility of assessing impaired intestinal permeability using confocal laser endomicroscopy as a potential target for IBD therapy.

This review focuses on the assessment of epithelial barrier function using confocal laser endomicroscopy and the utility to obtain healing of the intestinal permeability in patients with IBD in order to maintain prolonged clinical remission, increased resection-free survival and lower rates of hospitalization.

We used the PubMed and MEDLINE databases to identify articles about the use of confocal laser endomicroscopy in IBD to evaluate the permeability and barrier function of the intestinal epithelium in order to assess the degree of healing of the mucosa and predict relapses. Considering relevance to the scope of this review, we selected 51 articles that were published up to February 2023.

## 2. Intestinal Permeability-Definition and Methods of Evaluation

The term “intestinal barrier” was introduced in 2004 by Cummings et al. to describe a complex anatomical structure with a protective role that stands between the wall of the intestine and the luminal content [23]. The intestinal barrier function is the result of the connection of the epithelial cells by tight and adherent junctions. This tissue is constantly regenerated from stem cells. These cells originate at the base of the crypts, migrate to the tip of the intestinal villi or the surface of the colon, and are removed as they mature [24].

Permeability is a functional characteristic of the intestinal barrier, and according to Bischoff et al., is closely related to the intervention of the commensal microbiota and the elements of the immune system of the mucosa that prevent the luminal penetration of macromolecules and pathogens [25]. Alteration of the integrity of this barrier is specific to inflammatory bowel diseases and favors the translocation of macromolecules and pathogens into the blood. Moreover, increased permeability was also detected in approximately 20% of asymptomatic first-degree relatives of patients diagnosed with CD, which suggests the possible involvement of a genetic component [26].

IBD occurs as a result of a homeostasis imbalance between the host and the gut microbiome, triggering an abnormal immune response that induces damage to the integrity of the epithelial barrier [27]. The diversity of the intestinal microbiota, which include between 10 and 100 trillion microorganisms, forms a complex ecological entity that is dominated by Firmicutes, Bacteroidetes, Actinobacteria, Proteobacteria, and Verrucommicrobia phyla, responsible for energy substrates and immunomodulatory role on the nutrition, metabolism, and defense function of the human host [28].

Modern eating habits, based on a hypercaloric diet that has increased consumption of processed foods and is rich in proteins, carbohydrates, unsaturated fats or additives, combined with a low intake of vegetable fibers, can substantially influence the diversity of the intestinal microbiota which can cause the appearance of a dysbiosis associated with pathogenesis in IBD [29,30]. Fajstova et al. noticed that a high-sugar diet significantly increased *Escherichia coli* and *Candida* spp. and stimulated the formation of polymorphonuclear neutrophil infiltrates that disrupted the integrity of the intestinal barrier [31]. In another study, it was observed that a diet based on the consumption of increased amounts of glucose or fructose did not trigger the appearance of inflammatory responses, but significantly modified the intestinal microbiota by multiplying the bacteria that release mucolytic enzymes [32].

A significant number of methods were developed for the non-invasive assessment of intestinal permeability. The absorption of low molecular weight sugars, polyethylene glycols, or chromium-labeled Ethylenediaminetetraacetic acid (EDTA) was used in the 1980s on small groups of patients for the in vivo assessment of epithelial barrier function [33]. For example, sucrose is rapidly hydrolyzed to fructose and glucose by a digestive enzyme, which provides the possibility of estimating proximal gastric and duodenal permeability by measuring the amount of the substance in the urine [34].

Permeability, measured by the ratio between the urinary excretion of lactulose and mannitol 2 h after administration, reflects the differential absorption of large (paracellular pathways) and small (transcellular pathways) molecules in the small intestine. Due to the influence of the colonic microbiota on the metabolism of the two substances, they are not recommended for the evaluation of UC patients [35]. Sucralose is the only disaccharide that is not influenced by the action of the colon microbiota, and it can therefore be used to evaluate the permeability of the entire digestive tract [36]; the method’s performance is increased when sucralose is combined with other sugars (triple or quadruple sugar test). For example, the estimation of permeability in the stomach, small intestine, and colon was made based on the measurement at different time intervals of the urinary concentration of a solution composed of four sugars (lactulose, mannitol, sucralose, and sucrose) [37].

After oral administration of Technetium-99m diethylenetriaminepentaacetic acid (99mTc-DTPA), an important change in permeability was found in patients with CD and UC, with or without an active episode of relapse, in accordance with the degree of inflammation [38]. Jenkins et al. proved that oral administration of 51Cr-EDTA caused an increase in the permeability of the mucosa of the small bowel and colon. This concept was demonstrated by measuring the radiolabeled molecules in the urine after an oral administration, or by assessing the plasma clearance of the tracer [39]. However, if the administration of the substance was intrarectal, an increase in permeability was observed only in patients with CD [38].

Used mainly in experiments, Iohexol is a contrast agent that has a high molecular weight (821 Dalton) low intestinal absorption. Moreover, it is filtered glomerularly, so it does not bind to serum proteins without being influenced by secretion or tubular reabsorption [40]. A serum increase of this substance has been reported at 3 and 6 h after oral ingestion for 50% of CD patients and 31% of UC patients [41]. A much more detailed analysis could be performed by measuring the transmural electrical resistance and the unidirectional flow of 3H-mannitol. In this situation, the impairment of the barrier function is confirmed based on the decrease of epithelial resistance and the compensatory increase in intestinal permeability for 3H-mannitol [42].

Information regarding the integrity of the intestinal epithelium could be obtained by scanning the conductance using an electric current generated between two adjacent electrodes positioned on a luminal probe with multichannel intraluminal impedance testing. While this technique has been initially validated only for evaluating the integrity of the esophageal mucosa, recent studies have reported a low duodenal and jejunal impedance in patients with functional dyspepsia [43]. The analysis is performed using two electrodes embedded in a balloon inserted through the working channel of an endoscope. By inflating the balloon, the electrodes come in contact with the mucosa, and the impedance is recorded for 90 s [44,45]. Although radiological methods could provide in vivo information about the functional integrity of the intestinal barrier, they are not able to characterize the morphological changes that lead to altered permeability.

Using the possibility of in vivo microscopic analysis of the architectural and cellular details of the intestinal mucosa, CLE makes the transition between endoscopy and traditional histology [46]. CLE is a complex high-resolution technique that highlights the abnormalities of the cellular structures of the epithelium of the digestive tract by integrating a confocal laser microscope into the distal part of a conventional endoscope. Although there were two systems available at first, namely a probe as well as an endoscope-based CLE, only the probe remains available currently [18]. During the examination, a laser beam is used that generates an excitation wave of 488 nm; it can penetrating the mucosa up to a depth of 250 μm to obtain optical sections of 7 μm, which is the in vivo equivalent of histological images [47]. The laser beam is pointed towards the surface of the tissue and the light is reflected on a lens, which is then refocused in the same plane through a horizontal hole. This process allows the rejection of unfocused rays and produces magnified images up to 1000 times; the images can be stored digitally [48]. In combination with topical or intravenous administration of fluorescent dyes, CLE enables an optical biopsy for real-time histological diagnosis. This technique has proven its usefulness in the detection of changes in mucosal permeability, with the degree of severity interpreted according to the rate of elimination of epithelial cells, tight junction status, and extravascular leakage [49]. Sodium fluorescein 10% is frequently recommended, with intravenous administration of 5 mL and a maximum level of contrast being obtained approximately 3–5 min after bolus injection. Except for some minor adverse reactions (nausea, vomiting, hypotensive episodes, skin rashes at the injection site, and slight epigastric discomfort), a cross-sectional study that included 2272 procedures performed in 16 medical centers did not report any clinical events with significant impact after using fluorescein [48]. The disadvantage of using this substance is the need to rapidly collect the images as the quality gradually reduces due to the loss of tissue contrast, although the effect may last up to 30–45 min [50].

Topical application of acriflavine was used to assess the degree of cellular dysplasia by visualizing the cores and nucleoli in the structure of the surface cells of the epithelium. However, it has been withdrawn from medical practice due to its mutagenic and carcinogenic effects [50].

Currently, numerous endogenous proteins have been proposed as biomarkers for the direct or indirect assessment of the integrity of the intestinal barrier depending on the translocation of molecules normally present in the lumen of the digestive tract. Increased concentrations detected in the blood of various protein structural components suggest damage to intestinal permeability [51].

LPS binding protein (LBP) synthesized by hepatocytes represents an acute phase protein that is combined in the bloodstream with the lipopolysaccharides of Gram-negative microorganisms (LPS) released by translocation from the intestine. It is considered a marker of endotoxemia that indicates increased transepithelial absorption [51]. Having a similar structure to *Vibrio cholerae* enterotoxin, zonulin is an acute-phase reaction protein produced by the liver and various epithelia that controls the stability of tight junctions at the level of intestinal apical cells. It is considered one of the main factors that guarantee the good function of the intestinal barrier due to its effects on epithelial tightness. Increased concentrations of zonulin in blood or feces have been correlated with the important alteration of intestinal permeability in the case of diabetes, IBS, or IBD [52].

I-FABP (intestinal fatty acid binding protein) is a cytosolic protein present in differentiated enterocytes of the small intestine and in a reduced proportion in the colon. Under physiological conditions, there are reduced amounts in the blood. However, in the case of an altered intestinal barrier, specific for IBD, celiac disease, necrotizing enterocolitis or obesity, I-FABP accumulates in the bloodstream, reflecting an important microbial translocation that is considered a biomarker of the permeability of the digestive tract [34,53]. Calprotectin, the calcium and zinc-binding protein, represents approximately 60% of the soluble proteins of the granulocyte cytoplasm [54]. An elevated level of fecal calprotectin indicates migration of neutrophils to the mucosa or lumen of the intestine in case of barrier disorders. Due to complex stability and resistance to enzymatic degradation, calprotectin can be easily measured in feces and is regarded as one of the most sensitive biomarkers [55]. Despite these benefits, except for LPS and I-FABP, these biomarkers are not frequently recommended in medical practice to assess intestinal permeability because they are not validated by the results of studies conducted on large groups of patients (Table 1) [34].

## 3. CLE Scores for the Evaluation of Intestinal Permeability

Recent theories have aimed to expand the use of confocal laser endomicroscopy, with the orientation toward evaluating the function of the epithelial barrier. Various studies have defined criteria for the interpretation of CLE images based on the type of intestinal barrier dysfunction severity that was intended to be assessed [21,61]. Watson et al. designed a score to evaluate the level of alteration of the intestinal permeability depending on the epithelial microerosions and the intensity of the fluorescein signal detected in the intestinal lumen. Three categories were established: normal (Watson I score), with functional defects (Watson II score), or multiple microerosions in the lamina propria (Watson III score) [62]. Moreover, Kiesslich et al. reported that a value greater than or equal to 2 on the Watson score, consistent with fluorescein leakage, predicted clinical recurrence in the following 12 months in patients with IBD with a sensitivity and a specificity of 63% and 91%, respectively (*p* < 0.001) [21].

To establish the severity of the epithelial barrier dysfunction, a new quantitative score was developed, namely the Confocal Leak Score (CLS), which includes fluorescein leakage, epithelial cell loss, and cell junction enhancement in patients with mucosal healing based on the ratio between the number of images with one or more pathological characteristics and the total number of confocal images of the terminal ileum per patient, multiplied by 100, with values between 0 (absence of barrier dysfunction) and 100 (completely dysfunctional barrier). The CLS score has been validated as a measure of epithelial barrier dysfunction as it recognizes the physiological level of mucosal cell shedding, a characteristic that could be quantified by the Watson system [63].

The counting of epithelial gaps detected by CLE, with reference to 1000 cells from the intestinal villi, was proposed in 2011 by Liu et al. as an option for quantitative analysis of intestinal permeability. Although terminal ileum epithelial gap density in both CD and UC patients has been shown to be considerably higher than in asymptomatic controls, it has not been significantly associated with inflammatory disease activation. Moreover, these epithelial gaps have also been identified in the duodenum of UC and CD patients, suggesting that the entire gastrointestinal tract could be involved in both cases [61]. As the shedding of intestinal cells is a physiological regenerative phenomenon that can be followed by transient gap formation, the simple quantification of epithelial gaps cannot be adequate to identify patients with altered intestinal permeability. Other disadvantages refer to the laborious method of performing the procedure and the possible variability of interpreting the results [61].

By combining CLE features related to fluorescein leakage, crypt architecture, and microvascular changes, Chang-Qing et al. defined a score used primarily to evaluate inflammation in UC rather than intestinal permeability [64]. Buda et al. developed a score specific to laser confocal microscopy based on fluorescein leakage and crypt diameter designed to predict an episode of disease decompensation during 12 months of follow-up in patients with UC. Thus, a diameter of the intestinal crypts of more than 90 μm together with a pericryptic leak greater than 3100 pixels is correlated with a high probability of disease activity resumption [65].

In order to detect mucosal changes before and after the initiation of biological therapy, Hundorfean et al. developed the endomicroscopic mucosal healing score (eMH) assessed according to crypt numbers and their lumen deformity and vascular leakage, with values from zero to four. eMH presents high values of the performance parameters (sensitivity 100%, specificity 93.7%, and accuracy 94.44%), having the possibility of correlation with the histological (Gupta) or endoscopic (Mayo) scores [66].

Neumann et al. studied the CLE characteristics of inflamed mucosa in a group of 54 patients with Crohn’s disease and introduced an endomicroscopic activity score (CDEAS) that considers crypt distortion, the presence of microerosions, increased vascularity, the number of goblet cells and increased infiltration in the lamina propria to differentiate the non-inflammatory appearance of the mucosa in real time with an accuracy of 87% (Table 2) [67,68].

## 4. Impaired Intestinal Permeability Assessed by CLE—A New Potential Target for IBD Patients

CLE is considered a valuable imaging tool that has expanded diagnostic and treatment options for IBD. Compared to standard histology, it offers the advantage of dynamic analysis of the functional and morphological characteristics of the intestinal barrier [69]. The usefulness of this investigation lies in its capacity to assess epithelial permeability as well as the estimation of mucosal healing (MH) [70,71]. MH correlates with prolonged clinical remission, increased survival without surgical resection, and significantly lower hospitalization rates for patients with IBD [72,73,74].

Due to the benefit of direct observation of microvascularization in the lamina propria, CLE could stand as a useful tool for therapeutic guidance and prediction of relapses [75]. Several studies proved that the impairment of cell permeability and the function of tight junctions represented the key events involved in the pathogenesis of IBD, having a predictive role in the progression of the disease and the occurrence of relapse in the following 12 months [76]. For example, in the case of UC, it has been observed that clinical remission is associated with the presence of small, round crypts with a regular arrangement and an intact epithelial barrier, while in the active form of the disease, the endomicroscopic images highlight large crypts with a distorted appearance, a pronounced capillary vascularization in the lamina propria, and numerous microerosions that cause fluorescein leaks in the extravascular area [61,77].

In a study that included 47 patients with UC and 11 with CD, Kiesslich et al. proved the usefulness of CLE to quantify intestinal barrier dysfunction depending on the evolution of the epithelial cell elimination process. In patients in clinical remission, the results showed an increased rate of cellular excretion, accompanied by the presence of fluorescein leakage, which was associated with the development of relapse in the following 12 months. Moreover, the study concluded that the integrity of the epithelial barrier was reflected by the absence of signs of inflammation at the level of the mucosa, and the lack of leaks of fluorescein in the intestinal lumen [21].

Using this technique, Mace et al. investigated 12 patients with UC in clinical remission and with endoscopically normal appearance of the colonic mucosa, in whom there were identified signs of microscopic inflammatory activity that included abnormal crypt regeneration, accompanied by altered vascular permeability [78]. Compared to the subjective assessment of barrier function performed by Kiesslich et al., the authors obtained promising results using a software program recommended mainly in the setting of CLE and performed by less experienced endoscopists that allowed automatic quantification of fluorescein leakage and architecture of the crypts [23,79]. In a prospective study, Chang et al. focused on the investigation of intestinal permeability using a confocal leakage score to compare the evolution of symptomatic versus asymptomatic IBD patients. The results of the study stressed the fact that the intensification of clinical symptoms, expressed mainly in the form of pain and severe diarrhea, was correlated with the increase of intestinal permeability [63].

The advantages of using the CLE technique were also emphasized in a study by Karstensen et al. which included 50 patients with IBD, in the phase of clinical and endoscopic remission. The authors related the prediction of relapse to fluorescein leakage and microerosions [22]. However, Danish researchers used CLE to evaluate the longitudinal histological changes that occurred after the administration of various immunosuppressive therapies in UC patients. Although it was noticed that certain CLE characteristics (crypt sinuosity, distortion of crypt openings, and crypt density) were normalized after therapy, the results could not confirm the restoration of intestinal barrier integrity as no significant correlation could be proven between the colonic presence of fluorescein leakage and improvement in histopathological characteristics [80].

Four evaluations investigated whether relapse in IBD could be predicted by using CLE. In each study, the use of CLE was directed to areas of the gut that presented a normal endoscopic appearance [21,65,79,81]. In the case of patients with UC, Buda et al. were able to predict the occurrence of relapses by considering a score that combined the amount of fluorescein leakage and the diameter of the crypt that exceeded a selected limit value [65].

Similar results were obtained by Chang-Qing et al., who investigated a group of UC patients in clinical remission. Based on the use of CLE, the possibility of predicting relapse (sensitivity 64%, specificity 88.9% and accuracy 74.4%) was much higher for patients in whom signs of active inflammation (grade C or D) were detected compared to those with a much lower degree (normal appearance or chronic inflammation) [79].

In the study conducted by Turcotte et al., if the number of epithelial gaps determined by CLE exceeded the limit of 6/100 cells, this was considered a predictor of hospitalization and surgery in patients with IBD (*p* = 0.02). Moreover, an increase of only 1% in this number was found to be accompanied by a 1.10-fold increase in the risk of relapse (95% CI: 1.01–1.20) [82]. Although most researchers focused on assessing the alteration of the barrier in the lower intestinal tract, the results obtained by Lim et al. proved that confocal endomicroscopy could identify epithelial lesions that were not identified on conventional endoscopic evaluation in the duodenum of patients with UC and CD. Although the endoscopic appearance was normal, histological examination confirmed mild nonspecific duodenitis in 7 of 15 patients with CD, whereas no microscopic changes were detected in patients with UC [62].

Rath et al. assessed in an observational study the outcome of the disease depending on whether patients diagnosed with IBD reached endoscopic, histologic or restitution of the intestinal permeability during the initial colonoscopic and CLE evaluation. Regarding the group diagnosed with ulcerative colitis, from the category who achieved both endoscopic and histologic healing, a major adverse outcome was reported for 29.4%. In contrast, among patients with barrier healing confirmed by the CLE evaluation, the rate of adverse outcomes was 19.1% during the follow-up period. Of the patients included in the CD category with combined endoscopic healing and histologic remission, 51% reported at least a hospitalization during the study. In contrast, 29.6% of patients with established colonic barrier healing needed medical support. Moreover, none of the patients with intact intestinal permeability in the terminal ileum experienced a major adverse outcome [83].

Regarding the application of CLE in the pediatric field, Zaidi et al. compared the presence of epithelial gaps located in the duodenum in patients with IBD (16 CD and 10 UC) and without it (17 controls). It was observed that there was an increased number of epithelial gaps in the duodenum of IBD patients; this correlated with the presence of inflammation and CRP values only in the case of UC. This finding proved that the alteration of the epithelial barrier is a specific manifestation of IBD, not secondary to the intervention of inflammation. Later, the same group of children was examined using CLE to detect the presence of circulatory changes in the portions of the duodenum that were not affected by UC, but the results obtained did not correlate with inflammatory markers or disease activity [84,85].

Another study analyzed the value of the density of epithelial gaps detected by pCLE as a predictor of clinical relapse in a group that included 24 children diagnosed with IBD (13 CD and 11 UC). In the absence of inflammatory signs detected at endoscopy, it was demonstrated that an increased density of epithelial gaps at the level of the terminal ileum is a significant predictor for the occurrence of clinical recurrence in the following 13 months [86].

## 5. Conclusions

CLE is a potentially valuable tool that could expand the diagnostic and treatment targets in IBD. Moreover, the dynamic evaluation of the functional characteristics of the intestinal barrier presents a significant advantage over the histological examination, having the potential to select patients with a high risk of relapses. In addition to endoscopic healing and histologic or transmural remission, the recovery of the intestinal barrier function emerges as a possible target that could be included in future therapeutic strategies used for IBD.

## Figures and Tables

**Table 1 diagnostics-13-01230-t001:** Tests used in the assessment of intestinal permeability in IBD.

Procedure	Description	Area of Application	Localization of Intestinal Barrier Dysfunction	Benefits	Disadvantage	References
**Using chambers**	Measures transepithelial ion electrical resistance (TEER)	Investigate the dynamics of transcellular and paracellular permeability with molecules of different sizes	Different regions of the GI tract	Allows selection of the interested tissue	Laborious procedure Special training Limited availability of healthy human tissue	Vanuytsel et al. [34] Wallon et al. [56]
**CLE**	Assesses the incipient intestinal lesions using a laser beam (ʎ = 488 nm) that passes through the confocal aperture of a microscope included in the distal end of the endoscope	In inflammatory bowel diseases, confocal endomicroscopy combined with intravenous administration of fluorescent dyes, performs “*optical biopsies*” for real-time histological diagnosis based on the detection of epithelial gaps, fluorescein leakage into the intestinal lumen, and cellular elimination	Distal portion of the small intestine and colon.	Allows selection of the tissue of interest Evaluates the permeability and integrity of the intestinal barrier during the endoscopic examination Might offer the possibility to guide the therapy and predict relapse	Invasive test High equipment costs Training for the interpretation of the images and the use of the equipment	Vanuytsel et al. [34] Buchner et al. [50]
**^99m^Tc-DTPA** **^51Cr^ EDTA**	Measures the urinary excretion of radiolabeled chelates after oral administration	Assesses the impairment of intestinal permeability in patients with IBD regardless the activity status of the disease	The entire intestinal tract	Non-invasive test Resistant to bacterial degradation	The radioactive character limits the practical medical application Time-consuming (with longer urine collections up to 24 h)	Resnick et al. [57] Graziani et al. [58]
**Urinary sugars** **excretion**	Measuring the urinary concentration of monosaccharides /disaccharides at different intervals after administration	Evaluation of the mucosal barrier dysfunction depending on the absorption (paracellular and transcellular pathways) and excretion of different sugars	Gastric and duodenal segment (sucrose) Colon (sucralose) Small intestine (lactulose/mannitol) Whole intestine (triple or quadruple sugar test)	Non-invasive test Reduced cost of the procedure	Time consuming in order to collect the urine (up to 24 h) Individual variations due to the non-mucosal factors such as gastric emptying, intestinal transit, renal clearance Requires additional equipment (LC-MS) to detect low sugar concentration in the urine	Khoshbin et al. [59]
**Biomarkers**	Measurement of concentrations in plasma of LPS; LBP, sCD14, I-FABP, zonulin and in feces (calprotectin, and zonulin)	Evaluation of the dynamics of biomarker concentrations for the detection of bacterial translocation and alteration of the intestinal barrier	Different regions of the GI tract	Easy procedure to perform	Necessity of further studies in order to approve their usefulness for defining the intestinal permeability	Seethaler et al. [60]

^99m^Tc-DTPA—Technetium-99m diethylenetriaminepentaacetic acid; ^51Cr^ EDTA—chromium-labeled Ethylenediaminetetraacetic acid; LPS—lipopolysaccharide; LBP—LPS binding protein; I-FABP—intestinal fatty-acid binding protein; sCD14—soluble CD14; LC-MS—Liquid chromatography-mass spectrometry.

**Table 2 diagnostics-13-01230-t002:** CLE scores regarding inflammatory lesions and prediction of the recurrence of the disease.

Scores	Intestinal Segment Investigated/Type of Disease	Grading Systems	Application of the Score	References
**Watson scale**	Terminal ileum **UC/CD**	**I. Intact epithelial barrier:** with no fluorescein leakage **II. Functional defects:** shedding of single epithelial cells and visible fluorescein leakage into the intestinal lumen **III. Structural defects:** multiple microerosions in the epithelium; fluorescein signal detected in the intestinal lumen	Value greater than or equal to 2, had a sensitivity and a specificity of 63% and 91%, (*p* < 0.001) for prediction of relapse in the following 12 months.	Kiesslich et al. [21]
**Buda**	Colon **UC**	Evaluation of the images obtained with pCLE according to fluorescein leakage and crypt diameter	Predict an episode of decompensation of the disease during 12 months of follow-up in patients with UC (*p* < 0.001).	Buda et al. [65]
**Chang-Qing scale**	Colon **UC**	Evaluation of intestinal inflammation by analyzing the structure of intestinal crypts **A. Absence of inflammation:** Normal arrangement and dimensions of crypts **B. Chronic inflammation:** Crypts placed irregularly with enlarged distances between crypts **C. Acute inflammation:** The alteration of the architecture of the crypts is greater compared to B **D. Acute inflammation:** Severe destruction of the architecture of the crypts with/without abscesses	The ability to predict relapse had a sensitivity, specificity, and accuracy of 71%, 90%, and 79% for histologic features of acute inflammation, and 64%, 89%, and 74% for CLE criteria, respectively.	Chang-Qing et al. [64]
**Confocal Leak Score** **(CLS)**	Terminal ileum **UC/CD**	Assessment of intestinal permeability in each CLE image based on fluorescein leakage, epithelial cell loss and cell junction enhancement in patients with mucosal healing, with values between 0 (absence of barrier dysfunction) and 100 (completely dysfunctional barrier)	A score more than 13.1 was correlated with intestinal symptoms in IBD patients who reached mucosal healing having a 95.2% sensitivity and 97.6% specificity	Chang et al. [63]
**Endomicroscopic mucosal healing (EMH)**	Colon ** UC **	Score based on vascular leakage, number of crypts, their lumen deformity, with values between 0 and 4.	High values of the performance parameters (sensitivity 100%, specificity 93.7% and accuracy 94.44%, respectively) regarding mucosal changes.	Hundorfean et al. [66]

UC—ulcerative colitis; CD—Crohn disease.

## Data Availability

Not applicable.

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
