# Peer review of "Impaired Intestinal Permeability Assessed by Confocal Laser Endomicroscopy—A New Potential Therapeutic Target in Inflammatory Bowel Disease"

_diagnostics, 2023, doi:10.3390/diagnostics13071230_

Round 1
Reviewer 1 Report
According to manuscript for review the following comments may be helpful
1- introduction may included the reports of increasing prevalence after immigration from low incidence country to high incidence countries.
2the effects of immune modulators in intestinal barriers with more details :
Biomarkers in inflammatory bowel diseases insight into diagnosis..
Gasteroenterology and hepatology from bed to bench
3-Add more recent research for the role of microbiota and diet in permeability and absorption of intestinal wall in UC and CD.
4-How the CLE effects on the remission patients with IBD it seems must be added more research results.
Author Response
We would like to thank you for your constructive suggestions.
According to the manuscript for review, the following comments may be helpful:
- Introduction may include the reports of increasing prevalence after immigration from low incidence countries to high-incidence countries.
- The effects of immune modulators in intestinal barriers with more details: Biomarkers in inflammatory bowel diseases insight into diagnosis..Gastroenterology and hepatology from bed to bench
- Add more recent research for the role of microbiota and diet in permeability and absorption of the intestinal wall in UC and CD.
- How the CLE effects on remission patients with IBD it seems must be added more research results.
Response:
- Thank you very much for your suggestion. We have revised the manuscript and made the needed changes (Please see Page 2, Lines 45-58).
- Thank you very much for your suggestion. We have revised the manuscript and made the needed changes (Please see Pages 6-7, Lines 214-249).
- Thank you very much for your suggestion. We have revised the manuscript and made the needed changes (Please see Page 3, Lines 113-130).
- Thank you very much for your suggestion. We have revised the manuscript and made the needed changes (Please see Page 11, Lines 399-414).
Reviewer 2 Report
This is a narrative review article on CLE for IBD patients to evaluate its severities and potential for relapse. They also compared other modalities with CLE and classification scores for CLE. It is a well-written article. However, there are some concerns about this article, 1. Please summarize the article search methods in the introduction. 2. Please state the primary outcome, and secondary outcomes of this study, 3. The authors could present the figures of CLE with brief comments on their experiences.
Author Response
We would like to thank you for your constructive suggestions.
This is a narrative review article on CLE for IBD patients to evaluate its severities and potential for relapse. They also compared other modalities with CLE and classification scores for CLE. It is a well-written article. However, there are some concerns about this article:
- Please summarize the article search methods in the introduction.
- Please state the primary outcome, and secondary outcomes of this study.
- The authors could present the figures of CLE with brief comments on their experiences.
Response:
- Thank you very much for your suggestion. We have revised the manuscript and made the needed changes (Please see Page 3, Lines 92-96).
- Thank you very much for your suggestion. We have revised the manuscript and made the needed changes (Please see Page 2, Lines 88-91).
- Thank you very much for your suggestion. The device has been recently purchased and we did not have the opportunity to work on it. We are working on the protocol and we would like to use it as soon as possible.